# Emerging Nano/Micro-Structured Degradable Polymeric Meshes for Pelvic Floor Reconstruction

**DOI:** 10.3390/nano10061120

**Published:** 2020-06-05

**Authors:** Kallyanashis Paul, Saeedeh Darzi, Jerome A. Werkmeister, Caroline E. Gargett, Shayanti Mukherjee

**Affiliations:** 1The Ritchie Centre, Hudson Institute of Medical Research, Clayton 3168, Australia; kallyanashis.paul@monash.edu (K.P.); saeedeh.darzi@hudson.org.au (S.D.); jerome.werkmeister@hudson.org.au (J.A.W.); caroline.gargett@hudson.org.au (C.E.G.); 2Department of Obstetrics and Gynaecology, Monash University, Clayton 3168, Australia

**Keywords:** pelvic organ prolapse, mesh complications, nanofiber mesh, 3D printing, foreign body response, cell therapy, tissue engineering

## Abstract

Pelvic organ prolapse (POP) is a hidden women’s health disorder that impacts 1 in 4 women across all age groups. Surgical intervention has been the only treatment option, often involving non-degradable meshes, with variable results. However, recent reports have highlighted the adverse effects of meshes in the long term, which involve unacceptable rates of erosion, chronic infection and severe pain related to mesh shrinkage. Therefore, there is an urgent unmet need to fabricate of new class of biocompatible meshes for the treatment of POP. This review focuses on the causes for the downfall of commercial meshes, and discusses the use of emerging technologies such as electrospinning and 3D printing to design new meshes. Furthermore, we discuss the impact and advantage of nano-/microstructured alternative meshes over commercial meshes with respect to their tissue integration performance. Considering the key challenges of current meshes, we discuss the potential of cell-based tissue engineering strategies to augment the new class of meshes to improve biocompatibility and immunomodulation. Finally, this review highlights the future direction in designing the new class of mesh to overcome the hurdles of foreign body rejection faced by the traditional meshes, in order to have safe and effective treatment for women in the long term.

## 1. Introduction

Pelvic Organ prolapse (POP) is a debilitating gynaecological disorder that affects 1 in 4 women across all age groups. After menopause, POP affects over 50% of women who have given birth previously. Women with POP, face a 19% lifetime risk for the surgical intervention. Furthermore, the risk of reoperation is close to 30% [1]. Until recently, surgical reconstruction using non-degradable lightweight meshes was the most common treatment option for POP [2]; however, due to unacceptable post-surgical complications, these meshes have now been banned in Australia, New Zealand, USA and UK [3]. The direct cost of POP is significantly higher than $1 billion annually. As the elderly population is estimated to double by 2030, POP represents a significant health challenge for both clinical and biomedical engineering researchers [4,5,6]. Recent reports have highlighted a critical need to develop a new class of meshes that can provide a safe and effective treatment for women in the long-term. To address this conundrum, it is imperative that clinicians and researchers understand the multi-factorial scientific aspects of each other’s fields. This review highlights the clinical and material science aspects that directly impact the design and application of advancing polymeric meshes for treatment of POP. To this end, it is important to not only be aware of the emerging technologies, but to relate them to the clinical understanding of how POP progresses from childbirth to finally show symptoms at a later stage in life. Finally, we discuss the design attributes of emerging nano- and micro-structured meshes using fabrication techniques, namely, electrospinning and 3D melt-electrospinning, and their immunobiology with close attention to the biological events associated with foreign body response to these implants.

## 2. Pelvic Organ Prolapse and Non-Degradable Biomaterial Meshes

### 2.1. Aetiology of POP

POP is a caused by the herniation of the pelvic organs (uterus, bladder and/or bowel) into the vaginal cavity, due to weakened vaginal walls [7,8,9]. This results in a vaginal bulge or dragging sensation that can cause bladder, bowel, and sexual dysfunction to profoundly impact the quality of life of millions of women worldwide [10,11]. While vaginal birth is the predominant risk factor for POP [12,13,14], other risk factors include pregnancy, obesity and ageing [15]. Furthermore, management techniques of childbirth such as use of forceps add to the risk of developing POP. Common age-related gynaecological procedures such as hysterectomy further weaken the apical support of the pelvic floor, which can lead to POP. About 52.4% of patients undergoing hysterectomy will subsequently develop POP [16,17,18], which will likely require further surgical interventions.

### 2.2. Biomechanics of the Pelvic Floor during Childbirth

Pelvic organs, namely, uterus, urinary bladder and rectum, are located inside the pelvic cavity and supported by the pelvic floor muscles which are connected to the pelvic bones through connective tissues [19,20]. The pelvic floor muscles, comprising the puborectalis, pubococcygeus, and iliococcygeus [21], collectively referred to as the “levator ani muscle” (LAM), play a key role in maintaining and restoring pelvic floor function. Computer-based simulation studies have greatly increased our understanding of LAM mechanics, particularly maximum threshold stretching during normal physiology [21,22,23,24]. However, vaginal birth causes abnormal stress to the vaginal wall and LAM well beyond the critical limit of normal physiologic function that often results in tissue damage owing to non-elastic deformation. Vaginal birth comprises three stages of labour: cervical dilation (1st stage) involving uterine contraction, passage of foetus along the birth canal (2nd stage), and finally placenta expulsion (3rd stage). Childbirth is associated with high uterine contraction pressure increasing from 8 KPa to as high as 19 KPa [12,24,25,26], resulting in the dilation of the cervix that allows the fetal head into the birth canal called the “Pelvic inlet”, which measures 11 cm from the sacral promontory to the symphysis pubis. In this passage, the foetal head associates 7 cardinal movements along its journey through the birth canal towards vaginal introitus (Figure 1). As the journey progresses, the foetal head generates high pressure and stretches LAMs, specifically the medial portion of the pubococcygeus muscle, to a stretch ratio of up to 3.2, in contrast to the non-introitus stretch ratio of 2.17 (Figure 1) [26]. As a result, trauma may occur to pelvic organ structural supports, leading to an increase in the risk of biomechanical failure and collapse of the pelvic floor organs over time. This abnormal stretching causes both tissue and nerve damage in the pelvic floor, which leads to future POP.

### 2.3. Treatment Strategies for POP

Conservative treatment involving non-surgical methods is the first line of treatment and is recommended for POP. This treatment involves various lifestyle interventions such as weight loss, avoiding exacerbating activities (lifting, constipation and coughing). Additionally, strengthening the pelvic floor muscles can be achieved through pelvic muscle contraction by special muscle training (PFMT) at increased intra-abdominal pressures, endurance, timing of contractions and often electrical stimulation [27,28]. However, they cannot entirely prevent or treat POP, which necessitates surgical interventions (native tissue repair or mesh-based) for an estimated 20% of women [16]. When conservative measures fail, surgical intervention is the single most common procedure comprising traditional suturing with or without synthetic polymeric meshes. Reconstructive surgeries are highly patient specific and depend on many factors, like the location of anatomic defects, the severity of prolapse and symptoms, physical activity level, and the durability of repair [29]. Very often, gynaecological surgeons are challenged to select the appropriate surgery specific to the patient and weigh the benefits of native tissue repair with its high recurrence rate versus mesh-augmented repair with higher rate of post-surgical complications [30]. Several common gynaecological procedures include sacrospinous ligament fixation (SSLF) or uterosacral ligament suspension (USLS) anchoring the vaginal apex. Both SSLF and USLF have low risk potential but are limited by their relatively high recurrence rate [29].

### 2.4. Biomaterial Meshes in POP Reconstructive Surgeries

Due to the high failure rates of native tissue repair surgery, surgical repair of vaginal wall prolapse has been augmented using various synthetic meshes. Recently, surgical reconstruction has used various non-degradable polymers, particularly polypropylene (PP)-based macroporous lightweight monofilament commercial meshes, to augment pelvic floor reconstruction surgeries. Alongside PP meshes, other meshes are mono-/multifilament type formation and fabricated using either weaving or knitting processes (Figure 2) [31,32]. The different types of meshes appear with different filament types (Figure 2), and varying pore sizes and stiffness, such as: type I (macroporous PP meshes, >75 µm, Prolene, Gynemesh, Marlex and Restorelle (coloplast); Ultrapro; type II (microporous multifilament Polytetrafluroethylene (eTFE) meshes, <10 mm, such as Gore Tex); type III (macroporous multifilament meshes or with microporous, Vypro II, eTFE (Teflon), Surgipro, Mersilene, and Parietex); and type IV (hypo-microporous meshes, <1 mm) [19,31,32,33]. These were the first synthetic meshes used for POP and are synthesised with the presupposition of better tissue integration, increased collagen production, and with the aim of increasing tensile strength and elastic modulus of vaginal tissue. However, none of these meshes were designed to mimic extracellular matrix (ECM) like microstructure to facilitate better cellular migration, and therefore, better tissue integration. Biologic mesh material based on collagen scaffolds derived from donor sources was then used. Based on the donor source, the biologic meshes are divided into three groups, namely, autologous graft, allograft and xenograft [9,31]. Various sources include autologous dermis, fascia lata or rectus fascia, allograft cadaveric tissue, porcine intestinal submucosa, bovine pericardium [35,36]. They involve rigorous decellularisation steps to leave an organised collagen microstructured scaffold with histological similarity to the host tissue. These processes are costly, time consuming, produce inconsistent mechanical strength due to variable degradation rates, and increase the chance of disease transmission [37]. In contrast to synthetic meshes, biological meshes have limited availability, and potential risk of donor site morbidity. However, surgical reconstruction using these different types of meshes has resulted in variable clinical success, with about 11% of patients with POP surgery suffering from lifetime stress urinary incontinence (SUI) and 30% needing repeated surgeries due to failure of the graft. Surgeons face the challenge of choosing the type of mesh to be implanted and further need to determine how well they can augment native tissue repair. At present, there are no optimal methods, or computational, engineering or animal-based scientific data demonstrating an optimal treatment strategy for POP patients. Furthermore, surgery encounters unavoidable difficulties such as suturing complexity, intra-operative complications (bladder, urethra or rectal perforation) and perioperative complications (blood loss of approximately 500 mL to 1000 mL, urinary incontinence) [9,38].

### 2.5. Clinical Adversities and Ban of Transvaginal Meshes

Owing to the improved surgical outcome compared to the native tissue surgery, non-degradable transvaginal meshes were commonly used until recent times for POP treatment. However, following complaints and FDA warnings during the last decade, growing evidence has shown that the risk and adverse effects of these meshes outweighed its benefits. Unacceptable adverse events including the exposure/erosion [32,37] of mesh (Figure 3), pain, and infection occurred in up to 10–42% of surgeries, resulting in market withdrawal and banning of PP-based transvaginal meshes in Australia, New Zealand, the UK, and more recently in the USA [3]. The mismatch between the biomechanical properties of PP mesh and vaginal tissue impedes efficient tissue integration, which allows the mesh to move through tissue layers and become exposed (in vagina) or erode into other organs [39,40]. The intolerable adverse events such as vaginal extrusion, nerve entrapment, pelvic and sexual pain result the increase the rate of reoperation [39,40,41]. Ideally, transvaginal meshes used in vaginal repair should promote healthy neo tissue formation with minimal foreign body reaction. The commercial meshes lacked cues mimicking the ECM, which is crucial to restoring the tissue function [42]. The lack of biocompatibility also triggered undesired foreign body responses, and ultimately the failure of meshes requiring additional surgery to remove them from patients due to the unbearable nature of the complications. It is now known that such PP-based meshes disrupt the ECM [9,40] and smooth muscles [43] in the vaginal wall, leading to these detrimental adversities [2,44]. Therefore, it is now well understood that mesh materials need to be designed more carefully with more practical considerations to avoid these unwanted consequences [45,46,47]. The relevant factors include design, polymer choice, surface chemistry and, most importantly, the immune response it triggers within the body in order to control the tissue repair process in the long-term. At present, there is no optimal therapy or material that can treat POP, highlighting the urgent need for a new generation of surgical devices and biomaterials.

## 3. New Generation of Meshes for POP Surgery

### 3.1. Polymeric Materials in New Mesh Design

Emerging materials for POP treatment include mainly synthetic materials derived from a variety of biocompatible and biodegradable polymers such as polylactic acid (PLA) [48,49], polylactic-co-glycolic acid (PLGA) [50], polylactic-co-caprolactone (PLACL) [51], polyurethane (PU) [48,52], polycaprolactone (PCL) [53,54], and ureidopyrimidinone-polycarbonate (UPy-PC) [55]. These polymeric materials include knitted meshes and elctrospun meshes in both micro and nanofibrous form that have been functionalised using bioactive macromolecules such as animal origin collagen [56,57,58], gelatin [51,59], chitosan [60], silver (Ag) [61] and growth factors like fibroblast growth factor (bFGF) [62] and connective tissue growth factor (CTGF) [63] to improve biocompatibility. A growing body of evidence suggests that the new class of mesh design considers several attributes such as optimal stiffness, porosity, inflammatory response eliminating chronic foreign body reaction to avoid complications, and even surgical methods for their practical application. Recently, synthetic degradable polymers have been investigated and have shown promise as alternative to current POP meshes in various animal model. Designing an implantable construct for POP application involves the selection of polymers, as well as an optimum fabrication process to attain desired outcomes for augmenting the native tissue repair process. The selection of polymer for mesh design considers whether the fabricated construct can offer optimum fibre alignment [64], pore size, filament/fibre/strand thickness, sufficient mechanical strength to perform the implant surgery, controlled biodegradation/bioresorption to support tissue in growth [65,66] for extended periods of time and stiffness [35,36,37] matching the native tissue stiffness at the implant site. The biocompatibility of degradable meshes can be modified using plant-based aloe vera/alginate complex [53] as a printed top layer encapsulating a cell-based therapeutic agent. Tuning the biocompatibility of new mesh designs is pivotal to efficient integration into the surrounding tissue, ensuring a balanced and controlled inflammatory response that promotes minimal chronic inflammation and favours regeneration of healthy tissue rather than formation of a scar tissue [2,67]. Each of these aspects plays an important role in ensuring a normal tissue homeostasis is maintained leading to the required tissue healing.

### 3.2. Mesh Fabrication Using Electrospinning

Fabrication of a new generation of meshes has used nano and micro fabrication methods like solution electrospinning [68,69] and melt electrospinning [70,71]. Electrospinning is an attractive strategy as it enables the design of highly porous and ultrafine fibrous meshes with a large surface to volume topography (Figure 4). The different types of electrospinning set up include two jet electrospinning [72], cryogenic electrospinning [73,74], salt leaching electrospinning [74,75] and core-shell electrospinning [76] using a coaxial nozzle or a stable emulsion system (Figure 5). The fabrication parameters can vary porosity or gradient pore structure as per the requirement to achieve optimal surface roughness for promoting cellular adhesion and subsequent tissue integration. Electrospinning involves an electrohydrodynamic-based electrostatic force to draw a polymer fibre continuously, requiring three basic components: high voltage source, polymer solution ejecting from a metallic nozzle tip or needle and a collecting metal surface (Figure 4). Electrospun fibres have a remarkably large surface area to volume ratio, which can be as high as 103 times in the nanoscale compared to the micro-scale (Figure 4). The human vaginal tissue microenvironment consists of such ultra-fine architecture that allows normal cellular homeostasis [77]. Nanomaterials with such high surface areas mimic tissue ECM [78]. Electrospinning also allows researchers to tailor the shape, structure, design, surface properties and mechanical performance of the biomaterial such as stiffness and tensile strength based on the solution composition to achieve the desired biological effect [79,80,81,82,83].

Electrospinning, generally named solution electrospinning, has attracted substantial research interest for fabricating variable diameter polymeric fibres in the micrometre to nanometre scale in urogynaecology [51,55,63,69,77]. Despite the promising results of solution electrospun meshes in POP research, the potential limitations include the commercial scale-up of fabrication, hazardous solvent, limited tunability in the polymer concentration and repeatability of the construct. An alternative to solution electrospinning is forcespinning [84] (Figure 5) using centrifugal force instead of electrostatic force, which can offer a high production yield using versatile synthetic polymers at high and low concentration not limited to organic solvents and similarly, can produce a comparable fibre diameter range in micro- to nanometres [84,85]. This can be a potential alternative worth exploring for POP treatment.

### 3.3. Mesh Fabrication Using 3D Printing and 3D Bioprinting

In 1980, 3D printing was invented by Charles Hull, who named it “Stereolithography” and is synonymous with additive manufacturing (AM) or solid freeform technology (SFT) [86]. Despite widespread industrial applications, various limitations in traditional subtractive manufacturing resulted in the emergence of three-dimensional (3D) printing in regenerative medicine only as a recent innovation [87]. The integration of computer-aided design (CAD) modelling enables 3D printing to control a layer-by-layer additive process in a defined path, therefore, generating a customised, site-specific construct ensuring reproducibility in contrast to the traditional process. Primarily, the technology commenced as a cell-free printing for fabricating various dental implants, bone implants, eyeglasses and other implantable devices. Although 3D printing fabrication embraces various synthetic polymers such as polycaprolactone (PCL), poly L-Lactide Acid (PLLA), poly glycolic acid (PGA), poly Lactic-co-glycolic acid (PLGA) and others materials, including ceramics and metals, few synthetic polymers such as PCL and polyurethane (PU) are reported in medical research [54,87,88,89]. Unlike conventional processes like electrospinning, hydrogel synthesis and decellularisation of biological grafts, 3D printing enables significant control over pore size, geometry, interconnectivity, anisotropic morphology, all whist avoiding human interruption and physical contamination. From a clinical perspective, 3D printing of meshes can offer personalised meshes for POP surgeries as they can be specifically tailored to the patient tissue parameters that need repair.

Among the many alternatives of 3D printing processes, 3D melt electrospinning (MES) (Figure 6) is an electrical-mediated process similar to solution electrospinning but uses molten raw material instead of a solution. Despite having a similar electro hydrodynamic principle to solution electrospinning, both processes are quite distinct in viscoelastic behaviour affecting the net coulombic repulsion in the emerging droplet originated from the high voltage electric source [70,71]. The viscosity for solution electrospinning is significantly lower, resulting in lower capillary pressure compared to electrostatic pressure. This produces inherently randomised fibres and is inversely correlated for MES (higher viscosity causes higher capillary pressure), establishing a continuous molten jet stream that deposits onto the collector. MES holds a significant research interest owing to the lack of hazardous solvents and ease in automating the setup, resulting in a higher yield suitable for clinical application, even in clean rooms and GMP facilities. Despite leveraging these benefits, MES-based meshes are still in their infancy, and need further research to achieve fibre formation in the nanometre scale. Fibre diameter largely depends on the degree of balance between tangential electric and tensile stress due to the viscoelastic behaviour of the polymer melt [70]. Fine-tuning of these parameters to achieve optimal scaffolds for POP will require intensive engineering-based research before they can be considered for clinical application.

Since POP is a highly patient-specific anatomical condition, pelvic ultrasounds such as 3D endovaginal ultrasound (EVUS), 2D perineal pelvic floor ultrasound (pPFUS), transperineal ultrasound (TPUS), and translabial ultrasound (TLUS) [90,91] can potentially help identify the optimal size of implant required by the patient. The CAD model thus generated can be transformed into a transplantable 3D vasculature using 3D bioprinting (Figure 6), which might lead to future POP application. 3D bioprinting is a form of additive manufacturing that can be tissue engineered using a bio-ink, a highly biocompatible hydrogel carrier to encapsulate the cells to print layer by layer precisely, thereby constructing a 3D functional living tissue or artificial organ using a CAD model [92,93,94,95,96]. To date, bio printing technologies have evolved over time to produce systems such as inkjet, laser- assisted and extrusion-based bioprinting [96,97] (Figure 7).

Literally, “Bio-ink” refers to a printable biomaterial [98] comprising both natural compounds including polysaccharides, protein complexes, or synthetic polymers such as polyethylene glycol (PEG) and PLGA, and cells to produce an ECM-like micro/macro architecture material supporting cellular growth and functionalities (Figure 4) [98]. The choice of bio-ink must allow high-definition printability that largely depends on viscoelasticity, strong shear-thinning behaviour, ease of crosslinking, cell protection during printing, biocompatibility for cell proliferation and biodegradability enhancing tissue integration with native ECM [93,97]. The need to develop new classes of bio-ink is ever increasing; in particular to improve host mimetic vasculature, to achieve differentiation at the right time and place with the right phenotype. Incorporating cellular regulatory factors at the molecular, structural and physical level that will influence in vivo cellular dynamics is an increasing challenge for the science of biomaterial development. Hence, the synthesis of an optimal bio-ink tailored to the clinical application requires expertise in biology, engineering and chemistry. However, from a clinical perspective this is a knowledge gap worthy of research in order to use the benefits of 3D bio printing to address unmet clinical needs such as POP.

## 4. Practical Considerations in the Design of New-Generation Meshes

### 4.1. Controlling Foreign Body Response to Implanted Meshes

Following implantation of any foreign material, the immune system will be activated to respond to the implant. The Foreign Body Response (FBR) is a complex cascade (Figure 8), starting with adsorption of proteins to the material surface which leads to a provisional matrix formation that allows recruitment of neutrophil and mast cells to the injury site [77,99]. The material properties determine the amount and conformation of proteins that attach to the material surface and direct the immune system activation [100].

Following matrix formation on the material surface, the acute immune response begins following neutrophil infiltration. Activated neutrophils attempt to phagocytose and destroy the material by secreting proteolytic enzymes and Reactive Oxygen Species (ROS) and materials which are susceptible to oxidative environment undergo damage and corrosion in this early stage of FBR [102]. Following activation, neutrophils synthesise potent chemokines CCL2 and CCL4, which recruit monocytes, immature dendritic cells, macrophages and lymphocytes [103]. As monocytes infiltrate the wound, the acute phase (which usually lasts up to two days) resolves and neutrophils undergo apoptosis and are cleared from the implantation site [104]. Recruited monocytes differentiate to classically activated macrophages “M1” and produce a broad range of inflammatory cytokines including IL-1β, TNF-α, IL-6, chemokines and enzymes [105]. M1 macrophages remain at the implant site and secrete chemokines CXCL8, CCL2 and CCL4 that recruit more inflammatory cells and attempt to degrade the foreign material using secreted ROS enzymes [106].

Adherent macrophages differentiate to alternatively activated macrophages, “M2”, which express such Mannose receptor (MRC1) and secrete anti-inflammatory cytokines and molecules including IL-10 and Arginase I [107]. The continuous presence of macrophages and the overlapping transition of M1 to M2 phenotype switching lead to macrophage fusion and formation of multi-nucleated Foreign Body Gian Cells (FBGC) [101,108]. FBGCs are recognised as the hallmark of the chronic response to the implanted material and may result from the fusion of macrophages in attempt to phagocytose a large particle (Figure 9). However, their exact mechanism and role in FBR response is not fully understood.

During the chronic phase of the immune response, T lymphocytes are also activated and produce cytokines that modulate the pro-inflammatory and anti-inflammatory responses. Although the role of T lymphocyte in the chronic phase of the host response has not been fully elucidated, several reports have shown that T cell attachment to the material enhanced adherent macrophage/FBGC activation and cytokine production [108,109]. This sequential and orchestrated process results in release of PDGF, VEGF and TGFβ, which subsequently recruit fibroblasts [110,111]. Fibroblasts can appear from early to late days post mesh implantation and deposit collagen I and III in order to repair the damaged tissue. However, the excessive production of collagen (higher ratio of collagen I/III) can cause fibrotic encapsulation of the biomaterial, compromising its function [112]. It is these complex events of the FBR that determine the fate of the implanted device and modulating this response results in a successful outcome. Current research in the field therefore emphasizes improving the immunobiology of meshes by incorporating appropriate signalling molecules. These strategies involve surface property modifications and/or regenerative mesenchymal stem cell incorporation.

### 4.2. Biomaterial Design of Meshes to Reduce Foreign Body Response

Recent evidence suggests that promoting specific interactions between host tissues and the implant can impact the FBR. This has sparked significant interest in the development of degradable polymeric meshes including electrospun nanofiber mats and natural decellularised ECM-based meshes [51,113,114]. Irrespective of the choice of material, new-generation meshes need to apply fabrication strategies to proactively boost the immune system rather than display inert properties. Design parameters and the ultimate cues presented by biomaterials play a crucial role in modulating the response of host cells [115]. Of these, physical properties such as substrate stiffness, topography, pore size, and size of wear debris, and chemical properties including surface chemistry, ligand presentation and release of growth factors can be modified (Figure 10) to influence the behaviour of macrophages [116,117], the key cell involved in FBR. To date, clinical evidence from using commercial PP-based meshes suggests that heavier-weight mesh with lower porosity and higher stiffness exerts a profound negative impact on reconstructive surgeries, causing abnormal post-surgical complication and higher recurrence rate [9,42]. Although the mechanism of such post-surgical complications is poorly understood, recent studies in non-human primates have shown that such complications stem from undesirable and prolonged immune responses and may also be associated with decreased vaginal collagen, elastin, and smooth muscles [9,40]. Explanted meshes from women also suggested that sub-optimal meshes that lead to erosion and pain pique both the innate and adaptive immune systems of the body [19,118]. Furthermore, it is now becoming clear that even lightweight microporous stiff meshes are associated with exposures and erosions, with increased matrix metalloproteinase (MMP) activity, which leads to degradation of muscularis layer/smooth muscle content and an increase in the ratio of collagen subtypes III/I [42,45]. The same meshes may have highly successful results in hernia repair [34], but may fail in urogynaecology owing to the different and complex nature of vaginal tissue dynamics.

This is one of the key reasons for the banning of commercial meshes for vaginal surgeries, while they continue to be successfully used for hernia repair. Therefore, moving ahead, it is imperative to design materials specifically for their intended organ of use and to understand how they interact with the host’s immune system. It is also highly critical that the FBR elicited by meshes may drastically change with the alteration of a single parameter, such as polymer choice, fabrication method, porosity or macromolecule incorporation. For example, incorporation of RGD molecule to PEG hydrogel (RGD-PEG) resulted in a thinner and less dense fibrous capsule around the material compared with PEG [119]. Moreover, identifying the human plasma adsorbed protein on the surface of polymer MMA and PMMA revealed that they adsorbed different complement system proteins which modify the subsequent cell–material interaction and ultimately cause different foreign body response [120]. Hence, the new class of tissue engineered (TE) materials must be designed to guide the host response towards a healing and anti-inflammatory path unlike highly stiff PP meshes, which often showed acute and sustained inflammation leading to a final rejection [42,121,122].

### 4.3. Design Aspects for Promoting Tissue Integration

To avoid erosion, it is highly desirable that meshes for POP integrate well with native tissue through active vascularisation and induce sufficient biomechanical strength through physiologic cellular functioning and deposition of neo-tissue at the site. The major components of vaginal tissue are smooth muscle and ECMs, which are largely comprised of collagen (84%) and elastin (16%) [123]. The material composition of a TE construct should be designed to mimic both the native tissue composition and the architecture at the implant site. Such TE constructs potentially enhance the deposition of ECM proteins, collagen and elastin [123,124], in a guided way to replenish the damaged tissue site. The mechanical mismatch due to the difference in stiffness often results in stress shielding in solid materials, thus contributing an unexpected integration at the interface. Stress shielding is very common in a wide variety of physiological conditions in both hard and soft tissues [2,9,40]. The stiffer materials shield the adjacent tissue to shift the physiological loads through the interface of the new material. The imbalance in the physiological loads restricts the migration of the nutrients and results in a severe foreign body reaction. Therefore, highly stiff materials like PP mesh are detrimental to the normal tissue homeostasis between vaginal smooth muscle and ECM of the vaginal wall [40,125,126]. Hence, new material designs must embrace a biomechanical trade-off aiming for low-stiffness meshes, but with less tensile strength gained through a complex optimisation of fibre alignment, porosity and pore size, fibre diameter [50], biocompatibility, and with controlled biodegradation due to MMP activity [126,127]. It is desirable to have not only optimal material design, but also optimal tissue performance, including mesh integration with surrounding host environment in the long term. To augment native tissue repair, these important parameters need consideration for mesh to perform as a successful implant enhancing neo-tissue replacement. Low-stiffness and physiologically tuned degradable meshes, like degradable nanofiber meshes, have been fabricated to yield better results. The nanofiber meshes can potentially offer a similar nano scale range to native vaginal tissue that helps rebuilding the weakened vagina. The ability of such ultrafine meshes to mimic the native architecture promotes tissue regeneration of the damaged site to the native tissue and maintain homeostasis. Recent studies have elucidated the potential for ECM-mimicking meshes in the promotion of the infiltration of anti-inflammatory macrophages [51,63,66,77].

### 4.4. Incorporation of Therapeutic Stem Cells to Enhance Mesh Performance

Tissue engineering (TE) is an emerging field of research focusing on biomaterial-based scaffolds mimicking a native-like environment comprising target therapeutic cells and occasionally growth factors to improve tissue repair and regeneration [37,46,128,129]. Cell-based therapeutics appear to be a promising frontier, with the potential to cure a multitude of diseases and disorders [77,130]. To date, the primary focus of cell-based therapies for soft tissue regeneration has been to use adult multipotent mesenchymal stromal cells (MSCs) [131] due to their capacity for clonogenic expansion, high proliferative capacity or self-renewing rate, and differentiation into multiple lineages [132]. MSCs were first identified in bone marrow in 1976, and subsequently in other tissues including adipose tissue, dental pulp, umbilical cord, corneal stroma, cord blood, skeletal muscle, placenta and endometrium [133,134,135]. They have been observed to differentiate into many types of tissue and produce various angiogenic and growth factors that affect endothelial cell survival, induce tissue repair and inhibit apoptosis [136]. MSCs can also modulate the immune system through secretion of paracrine factors [137]. MSCs produce several mediators that affect the key players in innate and adaptive immune systems. MSCs inhibit T and B cell proliferation, and induce regulatory T cells via cell–cell contact or IDO and PGE2 synthesis [138]. Moreover, studies have reported a strong loop between M2 macrophage polarisation by MSC and adaptive immune systems. Activated T cells produce inflammatory cytokines like IFN and TNF, which increase COX2 and IDO in MSCs, which in turn induce M1 to M2 polarisation [139].

We found a small population of MSCs in the human endometrium with high proliferative and differentiation capacity [140]. Endometrial mesenchymal stem cells (eMSCs) are peri-vascular clonogenic cells that express typical MSC markers and differentiate into four mesodermal lineages [141]. We discovered their perivascular location and introduced SUSD2 as a single maker for eMSC isolation [142]. Furthermore, we found that eMSC expansion in serum-free media enriched with TGFβ receptor inhibitor maintained their stemness properties, i.e., prevented apoptosis and senescence and promoted their proliferation [143]. Unlike other tissues, human endometrium is a highly regenerative tissue, as demonstrated by its monthly cycles of regeneration, differentiation and shedding for over 400 cycles during a woman’s reproductive years [144,145]. For each cycle, the endometrium regenerates about 4–10 mm in thickness to perfection within a few days, producing scar-free neo-tissue. This enormous regenerative capacity of the endometrium is likely due to the presence of eMSCs [146]. Similar to MSCs, eMSCs exhibit adult stem-like properties including self-renewal, high proliferative capacity, and differentiation into multiple mesodermal lineages such as smooth muscle, osteocytes, adipocytes and chondrocytes [147,148,149]. The immunomodulatory effects of SUSD2+ eMSC are mediated by producing PGE2, TLRs, and cytokines [150,151]. Moreover, eMSCs also affect T cell function by suppressing ConA-stimulated murine T lymphocyte proliferation in a dose-dependent manner [152]. Furthermore, eMSCs can be easily obtained from post-menopausal women [148] in order to develop autologous cell-based constructs for surgery. Therefore, eMSCs are of particular of interest for tissue engineering of biomaterials to improve the outcomes during POP treatment.

## 5. Application of New Alternative Meshes for POP treatment

### 5.1. Nanostructured Meshes

In nature, tissue microenvironments, including that of vaginal tissue, are regulated by the nanoscale architecture of the ECM, which in turn helps to regulate the myriad biochemical signals in order to achieve cellular functions. Evidence has pointed towards alterations in this nanoscale architecture of the ECM in tissues from women with POP. The ultrafine biomimetic electrospun mesh, as mentioned earlier, can produce an ECM-like topography with a similar nanoscale architecture, thereby providing larger surface areas for adsorbing proteins and providing more binding sites for cell membrane receptors, unlike microscale and flat surfaces. The poly L-lactide-co-1-caprolactone (PLCL) is known to be biocompatible and has been studied by several groups for pelvic floor and vaginal tissue engineering applications owing to its elastic modulus, which closely matches that of vaginal native tissue. Biological materials are often blended with synthetic polymers to improve their tissue biocompatibility. Electrospun nanofibers of PLCL blended with macromolecules such as fibrinogen have been developed to reduce the FBR in terms of capsule thickness and fibrous tissue formation in a canine model of abdominal defect compared to polypropylene meshes [153]. However, PLCL/fibrinogen(Fg) blended nanofiber meshes showed higher angiogenesis in the short term only and not much difference in the long term. Yet, PLCL/Fg resulted in faster neo-vascularisation, better collagen fibre organisation and muscle regeneration compared with the polypropylene meshes, and did not erode at all [153]. Therefore, in order to develop successful surgical meshes for POP, it may be crucial to drive early neo-vascularisation processes to reduce the risk of deleterious FBR, erosion and other adversities. This is further supported by research from our team, which showed that macromolecule gelatin-based PLCL blended meshes not only reduced the pro-inflammatory FBR, but also significantly increased anti-inflammatory response in comparison with PLCL meshes alone [51]. We showed that blending of PLCL with gelatin significantly increased the hydrophilicity and pore size of the meshes. As a result, these blended meshes, despite having a pore size of less than 3 µm, could enable complete infiltration of therapeutic eMSCs in vitro [51]. This multifunctional aspect of the solution electrospinning enabled the nanofibrous mesh to become more hydrophilic, which results in better penetration of eMSCs. Such mesh properties were seen to have a significant impact on in vivo mesh performance, typically in terms of longer retention of cells at the site of implantation, site-specific secretion of collagen deposition, and higher ectopic cellular infiltration [51]. As a result, the FBR response was associated with more anti-inflammatory M2 macrophages. Delivery of therapeutic eMSCs also has a significant impact on the FBR and tissue regeneration process after mesh implantation. These improved meshes, together with therapeutic cells, is likely to yield the desired results in POP treatment. Even in the case of PLCL meshes without macromolecules, eMSC significantly upregulated the expression of angiogenic factors like *Vegfa*, *Ang1*, *Ctgf*, *Fgf1*, *Pdgfa*, *Tgfb1* and *Tgfb3* in vivo by 6 weeks in a mouse model [77]. In addition, eMSC-based tissue engineered nanofibrous PLCL meshes significantly downregulated several key acute inflammatory genes, namely *Tnfa*, *Il1b*, *Ccl4*, *Ccl5*, *Ccl7*, *Ccl12*, *Ccl19, Cxcl1*, *Cxcl-2*, *Ccr1* and *Ccr7*, and upregulated key anti-inflammatory genes like *Arg1* and *Mrc1* [77]. Furthermore, *Cxcl12* or *Sdf1a* gene was also upregulated, which suggests that eMSC enables modulation of FBR, angiogenesis and ECM regulation chemotactically. This was evidenced by the upregulation of genes associated with formation of neo-tissues and ECM regulation [77]. The expression of these genes collectively leads to the formation of newly synthesised ECM within the nanomeshes, and neo-vasculature in close proximity provides hope that meshes with desired biomechanical characteristics can be tailored in order to achieve the desired in vivo performance through tissue engineering (Figure 11). From a clinical perspective, this is a hallmark of a highly successful implant and could likely overcome the current hurdles faced in POP treatment.

Several other studies are leading towards the development of such nanofiber meshes using other highly promising materials such as PCL and PLGA to tailor them to patient needs [66,154]. In vitro studies have shown that fibroblasts from both POP and non-POP patients adhered, proliferated and produced ECM on degradable meshes [66]. The mechanical loading of such meshes in an in vitro setting with seeded patient cells showed that strain on meshes, which mimics a female pelvic floor, impacted the genes expressed by the POP patient cells [154]. Gentle cyclic straining upregulated genes involved in matrix synthesis (collagen I, III, V and elastin), matrix remodelling (α-SMA, TGF-β1, MMP-2) and inflammation (COX-2, TNF-α, IL8, IL1-β) [154]. Cells expressed relatively higher levels of mechano-responsive genes on electrospun scaffolds than on non-porous film, except for the inflammatory markers which expressed more strongly on the non-porous film. Collagen genes were expressed earlier under mechanical loading, and the ratio of I/III collagen increased. Matrix synthesis and remodelling were stronger on the electrospun scaffolds, while inflammation was more prominent on the non-porous film [154,155]. These finding, indicate that mechanical straining enhances the regenerative potential of fibroblasts for the regeneration of fascia-type tissues and limit the risk of scar tissue formation. Furthermore, these effects are stronger on an electrospun nanoscale texture. However, the best in vitro performance was on nylon, which is essentially a non-degradable polymer [66]. This further highlights the impact of fabrication and nanoscale architecture on mesh performance, with nylon having the thinnest fibre diameter of 117 nm compared to PLGA/PCL being 994 nm. Furthermore, nylon is also known to be more hydrophilic [156]. Nanomeshes of gelatin with a non-degradable polystyrene core that showed promising results to be considered for pelvic floor tissue engineering [155]. Thus, emerging research highlights that electrospinning provides a nanoarchitecture and improves cell–material integration for both degradable and non-degradable polymers [55,62], thereby amalgamating nanostructured mesh design along with cell therapy and tissue engineering, which is an attractive strategy for POP treatment.

### 5.2. Microstructured and 3D Printed Meshes

Microstructured electrospun meshes have been extensively explored using acellular, growth factor, and cell embedded forms in small and large animal models. Electrospinning is a highly versatile method that can produce meshes or tubular structures in nanofibrous as well as microfibrous formats [81,82]. Microfibrous degradable electrospun meshes have been shown to improve angiogenesis through controlled release of hormones. Oestradiaol-releasing degradable Polyurethane (PU) and PLA meshes both increased ECM production [52]. In particular, PLA meshes showed increased collagen I, collagen III, and elastin. Such meshes also significantly improved angiogenesis in an in vitro model, indicating them to be promising materials for use in pelvic floor repair and for improving the initial healing phase of a repair material following implantation. Microfibrous PLA meshes seeded with adipose-derived MSCs in a rodent model highlighted that such tissue engineered ultrafine meshes can improve the foreign body response and promote integration of the mesh in the host body [157]. Electrospun fibres find application in both acellular and cellular forms. Among these, a polyurethane (PU)-based microfibre mesh blended with estradiol [49,52] was fabricated to demonstrate the angiogenic potential using a chick chorioallantoic membrane assay [158]. Another study on PLA-based ascorbic acid-releasing electrospun mesh [159] demonstrated higher collagen deposition, and therefore, the potential to augment the native tissue repair in POP application. A comparative study using electrospun microfibre non-degradable PU and biodegradable UPy-PC, compared the tissue remodelling performance with non-degradable ultra-light-weight PP mesh [55]. The study demonstrated similar performance in tissue integration but showed a mild inflammatory response for both PU and UPy-PC mesh compared to the PP mesh. Other applications of electrospinning include the fabrication of core–shell nanofiber mesh encapsulating fibroblast growth factor (bFGF) and stem cell demonstrated promising results for POP application [63]. Other forms of electrospinning have been reported for the microsphere and nanosphere encapsulation of various therapeutic drugs, namely, electrospun PLGA microspheres to deliver simvastatin [160]. While small animal models are highly useful, preclinical studies using large animal models is an imperative step in developing novel therapies. In a sheep model of POP, the performances of two microfibrous electrospun scaffolds were evaluated and compared with clinical practice of native tissue repair and PP mesh surgery [55]. None of the meshes compromised vaginal wall contractility, and passive biomechanical properties were similar to those after native tissue repair. However, there was a 35% shrinkage over the surgery area for all surgical processes [55]. While all materials integrated well, exhibiting similar connective tissue composition, vascularisation, and innervation profiles, the inflammatory response was mild with electrospun implants, inducing both more macrophages, yet with relatively more type 2 macrophages present at an early stage than the PP mesh [55,157].

Macroporous PLCL scaffolds prepared by melt-extrusion have been shown to be biocompatible with vaginal stromal cells (SCs) and epithelial cells (ECs), and have therefore been applied for vaginal tissue engineering [161]. These scaffolds had pore sizes matching the stiffness of pre-menopausal women’s vaginas. The cells retained the viability, phenotype and morphology of vaginal epithelial and stromal cells in both separate and in co-culture conditions in vitro. Furthermore, gene expression of UPs, a group of transmembrane proteins that are expressed in urothelial cells showed that both vaginal ECs and SCs maintained their phenotype during the 14-day in vitro periods. 3D printed materials are increasingly gaining attention in tissue engineering, including vaginal repair and POP treatment. Recently, 3D printed thermoplastic PU meshes were designed and loaded with antibiotic levofloxacin in combination with fused deposition modelling to prepare safer vaginal meshes [162]. The printed meshes had lower stiffness than commercial PP meshes and exhibited significant bacteriostatic activity on both *Staphylococcus aureus* and *Escherichia coli* cultures [162], highlighting the potential of minimising the risk of infection from a POP surgical perspective. However, the in vivo performance of such meshes needs to be fully studied using suitable animal models prior to determining their potential in POP therapy. With the aim of developing an alternative therapy, we developed 3D printed PCL meshes and evaluated their in vitro and in vivo performance to demonstrate their potential in pelvic floor reconstructive surgeries. The tissue engineered scaffold, prepared by a two-step process involving melt electrospinning followed by 3D bioprinting of therapeutic cells in an aloe-vera-alginate hydrogel (Figure 12), exhibited a significantly lower FBR and degradation response compared to 3D printed meshes alone [53]. The bioprinted meshes with eMSC cells could recruit significantly higher numbers of host cell macrophages, and also promoted better integration with the host tissue (Figure 12). Our study clearly illustrates retention of therapeutic eMSC with MES mesh significantly lowered the pro-inflammatory and increased the anti-inflammatory wound-healing macrophages, thus highlighting the potential of MES meshes as a novel transvaginal mesh for POP treatment.

## 6. Future Directions

The ban on commercial non-degradable knitted meshes has left a devastating treatment gap for women suffering with POP, with no alternatives on the horizon. While there are significant scientific efforts towards developing alternative treatment options at both the nano and micro scales, translation of mesh requires engineers and scientists to work closely with clinicians. The key criterion of a successful alternative lies in a controlled design of meshes that combines appropriate biomechanical attributes in order to overcome mesh-tissue mechanical mismatch, promote favourable immune responses, and mimic the native ECM architecture. To this end, emerging fabrication processes such as solution electrospinning, melt electrospinning and 3D printing that embrace a computer interface to better control the fabrication steps and well as mesh architecture with minimal human intervention are desirable. It is likely that one or more of these fabrication methods may need to be combined to achieve all the desired properties of the mesh implant. While electrospun meshes have shown significant promise, offering structural and biomechanical cues for better cellular interaction and retention, further research is needed to improve their mechanical properties for feasibility in surgical application. Similarly, while melt electrospinning interfaces with a controlled computerised design, further research is required to extend its application to other promising polymers like PU, PLLA, PLACL. Furthermore, the therapeutic potential of MSCs, particularly eMSC, can be used in combination with biomaterials not only after menopause, but also soon after birth to prevent future POP. While there has been exponential advancement in the 3D bioprinting space, there is still a room for refinement of new bioink formulations that are compatible with cells, as well as suitable for electrospinning. Future studies are required to examine the combining of solution and melt electrospinning technologies to produce a composite architecture of nano- and micro-structured mesh fibres. Finally, the economic viability, combining the cost/benefits of these various emerging systems, needs to be tailored.

Apart from development of surgical devices, there is significant advancement in diagnostic devices that are developed through 3D printing as well as ultrasound methods for identifying POP and locating the weakened vaginal location. Pelvic floor ultrasound has emerged as a dynamic assessment tool for selecting appropriate patients for conservative or surgical management, thereby aiding in the counselling of patients on realistic expectations and strengthening the clinical examination of prolapse symptoms based on a current quantification known as POP-Q [90]. The diagnosis of levator defect after vaginal childbirth can be facilitated using pelvic floor ultrasound, particularly the identification of anatomical defect location. These new ultrasound technologies could enable the development of personalised mesh design based on a CAD generated from these images, thereby taking 3D printing technologies to the next level of sophistication. Computer-assisted robotic surgery [163,164] is already becoming a routine surgical method due to its precision and minimal invasiveness. In future, we envision a personalised mesh, designed using a computer interface integrated with a pelvic ultrasound, that can be implanted using computer-assisted robotic surgery.

## 7. Conclusions

POP is a multifaceted disorder that impacts millions of women worldwide. Although surgical reconstruction is the only solution for symptomatic POP, there are no optimal treatment options, with or without biomaterials. Following the ban of meshes, there is an urgent clinical need for the design of optimal, high-performance materials for alleviation of POP. To overcome the impediments of current meshes, it is imperative to design surgical constructs that mimic the properties of natural microenvironment and ultimately completely integrate with the host tissue. Such biomimetic mesh designs that degrade slowly over time are likely to not only prevent erosion of meshes but also eliminate deleterious foreign body responses and may be the most effective treatment therapy in the long term. In nature, cell behaviour and tissue structural development are supported by the nanoscale arrangement of the extracellular matrix (ECM) architecture that provides a larger surface area to adsorb proteins and present binding sites for cell membrane receptors. Although challenging, the current TE research understands that the emerging biomaterial in this field needs to mechanically match host tissue and mimic the host ECM environment to promote tissue regeneration and reduce foreign body reaction while also retaining therapeutic MSCs. Recent evidence suggests that highly stiff meshes are not the solution and degradable polymers with ultrafine morphology may be the path ahead to regenerate damaged vaginal tissue. Nanofabrication and 3D printing design strategies need to carefully consider biocompatibility and immunobiology of these constructs.

## Figures and Tables

**Figure 1 nanomaterials-10-01120-f001:**
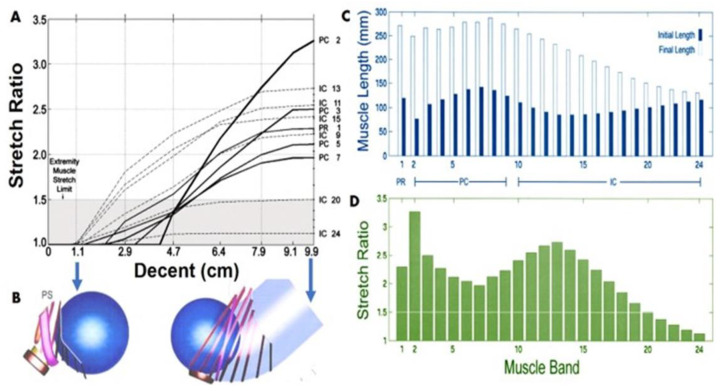
(**A**) The relationship between muscle stretch ratios in selected LAM, PC, IC and PR muscle bands and (**B**) foetal head descent in the birth canal. The shaded region denotes the values of stretch tolerated by non-pregnant appendicular striated muscle without injury. (**C**) Initial and final muscle length. (**D**) Maximum corresponding stretch ratio of each LAM muscle corresponding to foetal head descent. Reproduced with permission of [26]. Copyright NIH Public Access, 2005.

**Figure 2 nanomaterials-10-01120-f002:**
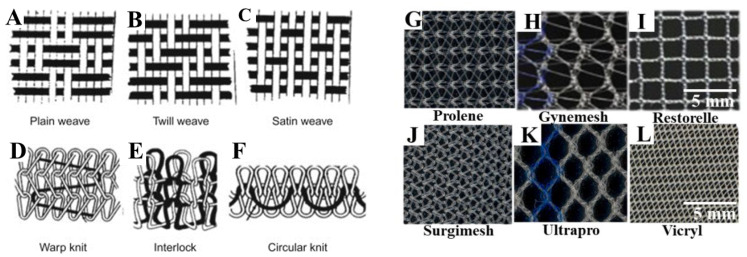
Mesh topography showing fabrication patterns of various types of weaves (**A**–**C**), types of knits (**D**–**F**) and commercially available meshes (**G**–**L**) in POP surgery. Reproduced with permission of [19]. Copyright Elsevier, 2016. Reproduced with permission of [34].

**Figure 3 nanomaterials-10-01120-f003:**
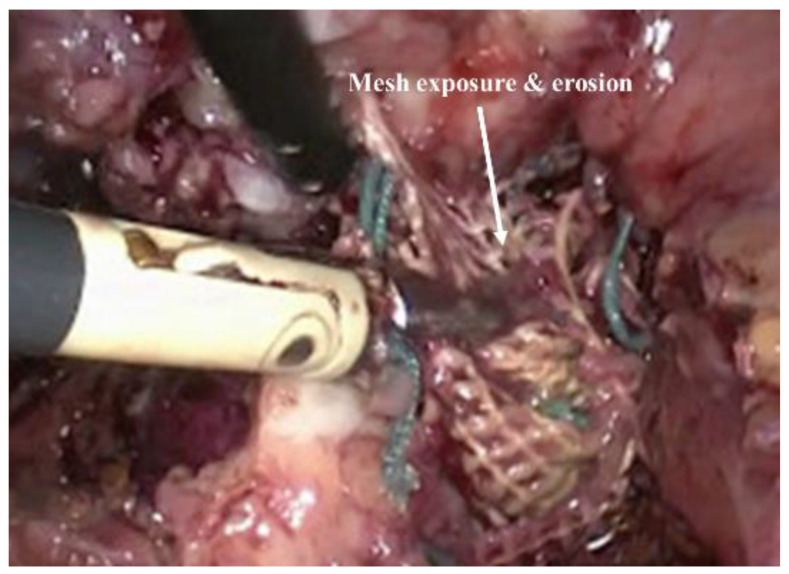
Mesh exposure and erosion of using non degradable meshes in POP surgery. Reproduced with permission of [39]. Copyright AAGL, 2017.

**Figure 4 nanomaterials-10-01120-f004:**
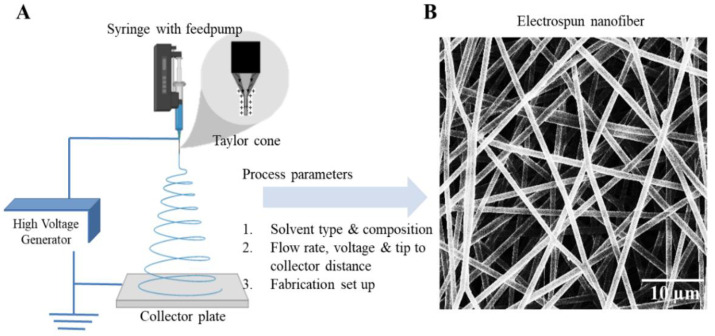
(**A**) Electrospinning setup and optimisation parameters. (**B**) Electrospun nanofiber. (**C**) Schematic showing the nano scale interaction with the cell. Reproduced with permission of [78]. Copyright AAAS, 2005.

**Figure 5 nanomaterials-10-01120-f005:**
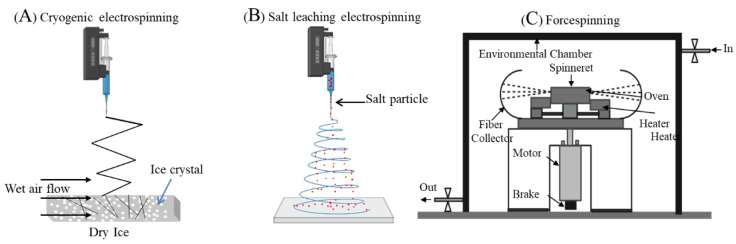
Different types of electrospinning fabrication setups (**A**,**B**). Forcespinning setup (**C**). Reproduced with permission of [74]. Copyright Elsevier, 2016. Reproduced with permission of [84]. Copyright Elsevier, 2010.

**Figure 6 nanomaterials-10-01120-f006:**
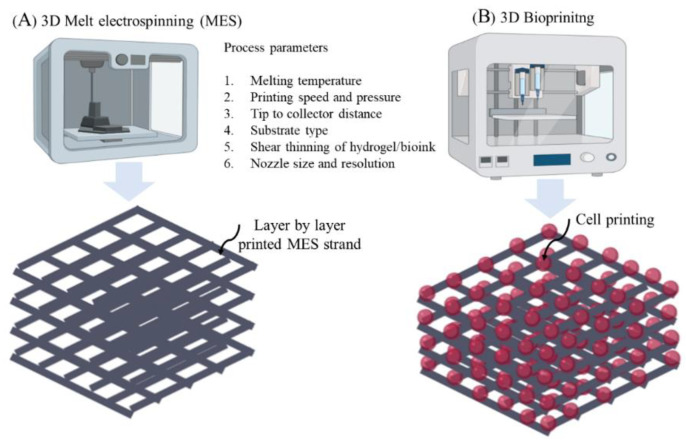
3D printing setups showing (**A**) 3D Melt electrospinning, and (**B**) 3D Bioprinting.

**Figure 7 nanomaterials-10-01120-f007:**
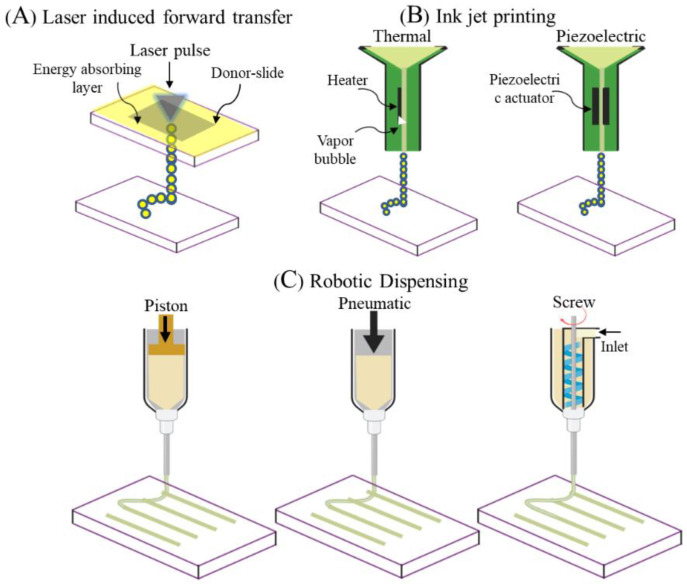
Different types of electrospinning fabrication setups (**A**,**B**). Forcespinning setup (**C**). Reproduced with permission of [93]. Copyright Wiley-VCH Verlag GmbH & Co. 2013.

**Figure 8 nanomaterials-10-01120-f008:**
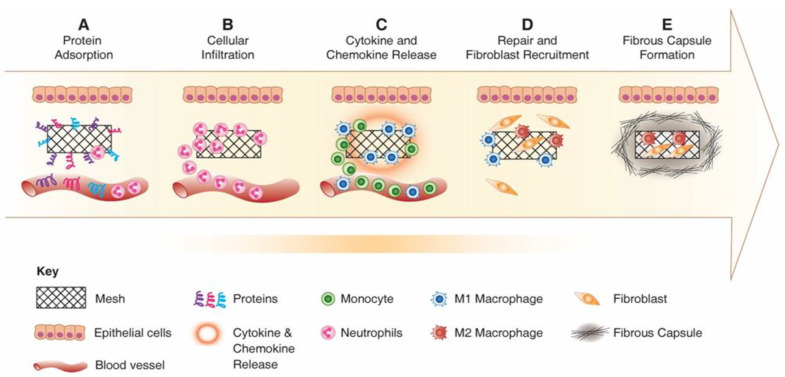
Schematic showing the foreign body response to an implanted inert biomaterial in the host’s body. (**A**) Protein adsorption; (**B**) cellular infiltration and acute inflammation; (**C**) chronic inflammation, cytokine release and further cell recruitment; (**D**) fibroblast recruitment and collagen matrix deposition; (**E**) formation of fibrous capsule. Reproduced with permission of [101].

**Figure 9 nanomaterials-10-01120-f009:**
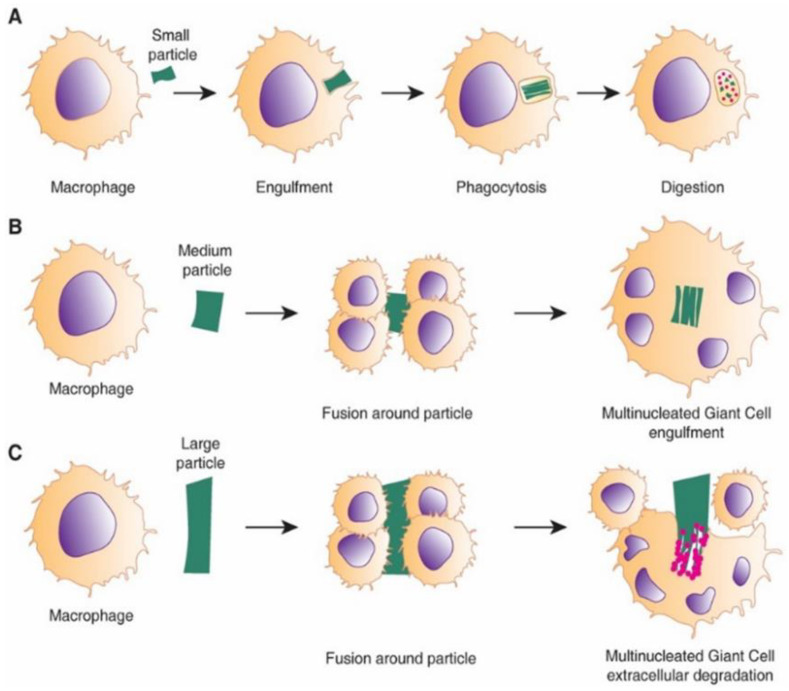
Schematic showing the process of FBGC formation by macrophages responding to foreign particles of different sizes. (**A**) Phagocytosis; (**B**) multinucleated FBGCs around the particle; (**C**) multiple FBGCs attempt to fuse around the larger particle causing extracellular degradation. Reproduced with permission of [101].

**Figure 10 nanomaterials-10-01120-f010:**
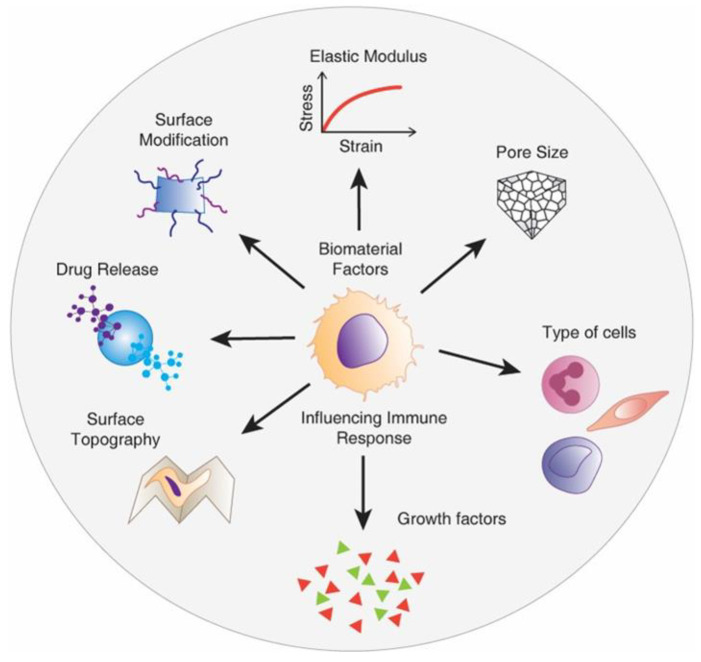
Schematic showing material design factors influencing the macrophage-mediated foreign body response to biomaterial implants including pelvic floor reconstruction. Reproduced with permission of [101].

**Figure 11 nanomaterials-10-01120-f011:**
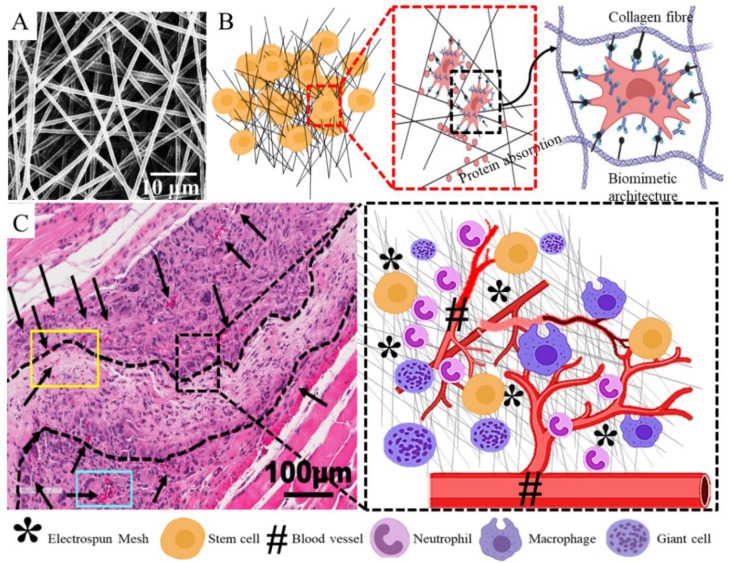
Tissue engineering attributes of electrospun nanofiber mesh showing nanofiber mesh (**A**), cell seeding and interaction with nano scale feature (**B**), tissue integration without formation of fibrous capsule and host response toward tissue healing (**C**). Reproduced with permission of [77]. Reproduced with permission of [78]. Copyright AAAS, 2005.

**Figure 12 nanomaterials-10-01120-f012:**
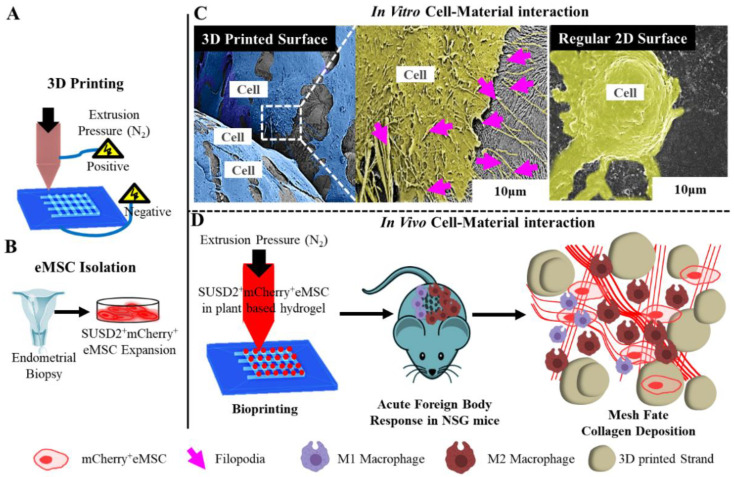
Tissue engineering attributes of 3D printed surface showing 3D melt electrospinning (**A**), Endometrial stem cell (eMSC) isolation using unique marker SUSD2 and transduction of mCherry gene (**B**), the difference in cytoskeletal morphology of migrating cells (**C**), 3D printing and bioprinting combination toward tissue healing in NSG mice (**D**). Reproduced with permission of [53]. Copyright Elsevier, 2019.

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
