# Peer review of "Emerging Nano/Micro-Structured Degradable Polymeric Meshes for Pelvic Floor Reconstruction"

_nanomaterials, 2020, doi:10.3390/nano10061120_

Round 1

Reviewer 1 Report

I have read with interest article entitled: „Emerging nano/micro-structured degradable polymeric meshes for pelvic floor reconstruction”. In this manuscript Authors described current approach and future perspective for POP treatment using polymeric meshes. Within the article Authors included extensive knowledge from the field of polymer meshes production, properties and application. In my opinion, in order to make manuscript more reader-friendly it could be shortened by electrospinning and 3D printing methods description and chapter “Controlling Foreign Body Response to Implant Meshes”. These are additional information without which manuscript will not lose its value. In order to improve manuscript value some minor changes have to be performed:

  1. Page 4 line 135 – “porcine intestinal submucosa” was duplicated
  2. Page 5 line 170: – “such as” was duplicated
  3. Page 12 line 414 – two dots after vagina
  4. Page 12 line 424 – according to International Society for Cell & Gene Therapy shortcut MSC should be expanded as a Mesenchymal Stromal Cells rather than Mesenchymal Stem Cells
  5. Page 17 line 627 – references in the upper index

Author Response

We are thankful for the valuable review. Please find our response to reviewer attached here. 

Reviewer 2 Report

I appreciated the present review for the relevance of the topic and the clarity of the discussion. Furthermore, the description of the different phases of the immunosystem reaction to a foreign body is very interesting because the knowledgesabout it are constantly evolving.

Only some inaccurancies, easily editable:

  • Lines 134-135 (twice the same phrase);
  • Line 165 (mimcking);
  • Line 179 (such as);
  • Lines 484-486;
  • Lines 489-494 ;
  • Lines 580-583 (what does it mean?)
  • The acronym ECM has been not named the first time that is reported the extracellular matrix.

In my opinion this work could be accepted after this minor, formal revisions.

Author Response

(The authors gave the same response as above.)
